# Safety of ChAdOx1 nCoV-19 vaccination in patients with end-stage renal disease on hemodialysis

I-Ning Yang[1]☯, Chin-Li Lu[2]☯, Hung-Jen Tang[3], Yu-Chi Kuo[4], Li-Hwa Tsai[4], Kuan Chieh Tu[5], Jhi-Joung Wang[6], Chih-Chiang Chien [1]*

1 Department of Nephrology, Chi-Mei Medical Center, Tainan, Taiwan, 2 Graduate Institute of Food Safety, College of Agriculture and Natural Resources, National Chung Hsing University, Taichung, Taiwan, 3 Department of Infectious Diseases, Chi-Mei Medical Center, Tainan, Taiwan, 4 Department of Internal Medicine, Nephrology Division, Chi-Mei Hospital, Chiali, Tainan, Taiwan, 5 Department of Internal Medicine, Chi-Mei Medical Center, Tainan, Taiwan, 6 Department of Medical Research, Chi-Mei Medical Center, Tainan, Taiwan

☯ These authors contributed equally to this work.
* ccchien58@yahoo.com.tw

**Data Availability Statement:** All minimal underlying data set files are available from the DRYAD public repository (DOI: 10.5061/dryad. 47d7wm3h3).

## Abstract

### Background

COVID-19 vaccination is essential. However, no study has reported adverse events (AEs) after ChAdOx1 nCoV-19 vaccination in patients with end-stage renal disease (ESRD) on hemodialysis (HD). This study investigated the AEs within 30-days after the first dose of ChAdOx1 nCoV19 (Oxford-AstraZeneca) in ESRD patients on HD.

### Methods and findings

A total of 270 ESRD patients on HD were enrolled in this study. To determine the significance of vascular access thrombosis (VAT) post vaccination, we performed a self-controlled case study (SCCS) analysis. Of these patients, 38.5% had local AEs; local pain (29.6%), tenderness (28.9%), and induration (15.6%) were the most common. Further, 62.2% had systemic AEs; fatigue (41.1%), feverishness (20%), and lethargy (19.9%) were the most common. In addition, post-vaccination thirst affected 18.9% of the participants with female predominance. Younger age, female sex, and diabetes mellitus were risk factors for AEs. Five patients had severe AEs, including fever (n = 1), herpes zoster (HZ) reactivation (n = 1), and acute VAT (n = 3). However, the SCCS analysis revealed no association between vaccination and VAT; the incidence rate ratio (IRR)-person ratio was 0.56 (95% CI 0.13–2.33) and 0.78 (95% CI 0.20–2.93) [IRR-event ratio 0.78 (95% CI 0.15–4.10) and 1.00 (95% CI 0.20–4.93)] in the 0–3 months and 3–6 months period prior to vaccination, respectively.

### Conclusions

Though some ESRD patients on HD had local and systemic AEs after first-dose vaccination, the clinical significance of these symptoms was minor. Our study confirmed the safety profile of ChAdOx1 nCoV-19 in HD patients and presented a new viewpoint on vaccine-related

**Funding:** This study was supported by grants CMFHR11002 from Chi-Mei Hospital. The funders had no role in study design, data collection and analysis, decision to publish, or preparation of the manuscript.

**Competing interests:** The authors have declared that no competing interests exist.

AEs. The SCCS analysis did not find an elevated risk of VAT at 1 month following vaccination. Apart from VAT, other vaccine-related AEs, irrespective of local or systemic symptoms, had minor clinical significance on safety issues. Nonetheless, further coordinated, multi-center, or registry-based studies are needed to establish the causality.

## Introduction

End-stage renal disease (ESRD) dialysis patients have distinct characteristics, including in-center hemodialysis (HD), old age, multiple comorbidities, and immune dysfunction [1,2]. The coronavirus disease 2019 (COVID-19) is associated with higher infection risk and subsequent morbidity and mortality in ESRD dialysis patients [3]. The relative risk of mortality was 45.4 for in-center HD patients with COVID-19 as compared with the general population [4]. Historically, this specific population had a higher mortality rate after pandemic viral infections, such as influenza (OR: 1.26, 95% CI: 1.15–1.38) [5]. Prioritizing these patients for vaccination is essential. However, most preexisting vaccine trials enrolled primarily healthy adults and explicitly excluded individuals with kidney disease [6]. ESRD patients generally have a blunted vaccination response due to uremia-associated immune dysregulation, as demonstrated with vaccines against hepatitis B, influenza, and pneumococcus [7–9]. Hence, this population may require a distinct vaccine strategy. To date, only a few published studies have evaluated the safety and immune response of mRNA vaccines (BNT162b2 and mRNA-1273) in patients on dialysis [10–12].

Considering that the ChAdOx1 nCoV19 (Oxford-AstraZeneca) vaccine has a different vaccine platform from the mRNA vaccines, its vaccine adverse events (AEs) might be different from those of mRNA vaccines. Moreover, ESRD patients on HD are distinct from the general population and they require vascular access for dialysis. No study has reported the incidence of AEs following vaccination with the first dose of ChAdOx1 nCoV19 in ESRD patients on HD.

In Taiwan, a new wave of the COVID-19 pandemic started abruptly in May 2021. Taiwan did not own enough vaccines at that time. Under the national policy, Oxford-AstraZeneca was the only available vaccine for high-risk populations, including healthcare workers and dialysis patients. Taiwan CDC and Taiwan Society of Nephrology strongly promoted vaccinations among ESRD patients to deter further spreading. Therefore, we conducted a study to investigate the local, systemic, and severe AEs in ESRD patients on HD during the 30-day follow-up period after the first vaccination with ChAdOx1 nCoV19 (Oxford-AstraZeneca).

## Methods and methods

### Study design and participants

This retrospective medical chart review study, which enrolled adult patients with ESRD on maintenance HD. All participants had tested negative for COVID-19 PCR before receiving the first dose of ChAdOx1 nCoV19 (Oxford-AstraZeneca). This study was conducted in compliance with the Declaration of Helsinki and was approved by the Research Ethics Committee of Chi Mei Hospital (IRB No. 11006–010). In addition, the requirement of patient informed consent was waived.

We recorded the clinical assessment with a questionnaire for vaccine safety on days 2 and 7. In the questionnaire, we defined localized AEs (injection site induration, itch, redness,

tenderness, warmth, swelling, and pain), systemic AEs (chills, fatigue, headache, muscle ache, joint pain, lethargy, insomnia, nausea/vomiting, diarrhea, abdominal pain, difficult breathing, chest pain/tightness, feverishness [defined as body temperature >37˚C and <38˚C], fever [defined as body temperature ≥38˚C], any fever [defined as body temperature >37˚C], anorexia, and thirst), and warning symptoms for unusual thrombotic events and thrombocytopenia [13]. We recorded each patient's smoking habits and paracetamol use and any unexpected visits to the emergency department (ED) or hospitalization within 30 days after vaccination. For unsolicited AEs, medical charts were reviewed by two clinicians to evaluate the potential causality. Once possible severe AEs were documented, a self-controlled case series (SCCS) analysis was conducted to verify the association between the vaccination and the event.

Important baseline comorbidities for ESRD patients on HD, including diabetes mellitus (DM), cardiovascular disease (CVD), and chronic obstructive pulmonary disease (COPD), were collected. We retrieved baseline laboratory tests, including albumin level and platelet count, before the first dose of the ChAdOx1 nCoV19 vaccine. We presented the dialysis dosage by Kt/V, calculated automatically by the computer system.

## Statistical analyses

We compared the baseline characteristics and post-vaccination incidence of AEs between patients aged ≤ 55 years and those > 55 years in our ESRD cohort. The incidence of individual symptoms of local and systemic AEs was also compared between days 2 and 7 using McNemar's test. To investigate the relationships between risk factors and incidence of each local or systemic AE within 2 days after vaccination, we used a multivariable binary logistic regression model to estimate the adjusted odds ratio (OR) and 95% confidence interval. All baseline characteristics listed in Table 1 included in multivariate analyses except analgesics use and type of vascular access. The reasons for not including the two factors were that: (1) analgesics use may mediate the effect of vaccination on complications and (2) almost all patients under 55 years-old have an AVF rather than other types of vascular access. We then excluded habitual smoking, history of COPD, and CVD in the models. It was because the very low prevalence and extremely uneven distribution of smoking and COPD, which resulted in unstable estimates of regression coefficients. CVD was excluded due to its non-significant association with all listed symptom in logistic regression models. We found that the patterns of significant associations were quite similar in regression models with and without inclusion of COPD, smoking, and CVD; considering a better efficiency, we present our final models with exclusions of these three variables.

Additionally, we carried out a self-controlled case series (SCCS) analysis of incident vascular access thrombosis (VAT) events following the first-dose ChAdOx1 nCoV-19 vaccine [14]. The incidence rate (IR) of VAT events was calculated in three time periods: 0–3 months before vaccination (pre1), 3–6 months before vaccination (pre2), and the first-month post vaccination (post). We estimated IR by dividing the number of persons (IR-person) or the number of events (IR-event) by observed person-months. IRs in pre1 and pre2 were compared with that in post-period using a conditional Poisson regression model. p-values ≤ 0.05 were considered statistically significant. All data were analyzed using SAS statistical software (SAS System for Windows, Version 9.4, SAS Institute Inc., Cary, NC, USA).

## Results

### Baseline characteristics in our ESRD study cohort

A total of 270 ESRD patients on HD were enrolled (Table 1). Our ESRD cohort tended to be older and had a low BMI and multiple comorbidities compared to the general population.

**Table 1. Baseline characteristics and adverse effects on day 2 of ChAdOx1 nCoV-19 (Oxford-AstraZeneca) vaccination (first dose) in ESRD patients on hemodialysis (N = 270).**

| Baseline characteristics and adverse effects 2 days after vaccination | | Overall | | Age (years) | | | | P-value |
|---|---|---|---|---|---|---|---|---|
| | | | | ≤ 55 (n = 47) | | >55 (n = 223) | | |
| Baseline characteristics | | | | | | | | |
| Age, years | | **66.4** | (11.5) | **48.6** | (5.69) | **70.1** | (8.53) | <0.001[a] |
| Sex | female | **107** | (39.63) | **10** | (21.28) | **97** | (43.50) | 0.025 |
| | male | **163** | (60.37) | **37** | (78.72) | **126** | (56.50) | |
| Weight, kg | | **61.7** | (12.80) | **68.0** | (16.50) | **60.4** | (11.50) | 0.004[a] |
| BMI, kg/m$^2$ | | **23.3** | (3.59) | **23.9** | (4.25) | **23.2** | (3.44) | 0.257[a] |
| | <24 | **173** | (64.07) | **28** | (59.57) | **145** | (65.02) | 0.479 |
| | ≥24 | **97** | (35.93) | **19** | (40.43) | **78** | (34.98) | |
| Albumin, g/dL | | **3.73** | (0.39) | **3.84** | (0.41) | **3.71** | (0.38) | 0.038[a] |
| Kt/V | | **1.74** | (0.35) | **1.61** | (0.39) | **1.77** | (0.34) | 0.004[a] |
| Platelet, 10$^3$/uL | | **183** | (62.10) | **200.0** | (65.20) | **179.4** | (60.90) | 0.038[a] |
| Diabetes | | **149** | (55.19) | **19** | (40.43) | **130** | (58.30) | 0.025 |
| CVD | | **183** | (67.78) | **27** | (57.45) | **156** | (69.96) | 0.095 |
| COPD | | **43** | (15.93) | **3** | (6.38) | **40** | (17.94) | 0.050 |
| Smoking | | **23** | (8.52) | **10** | (21.28) | **13** | (5.83) | <0.001 |
| Use analgesics | | **94** | (34.81) | **23** | (48.94) | **71** | (31.84) | 0.025 |
| Types of vascular access | AVF | **200** | (74.07) | **41** | (87.23) | **159** | (71.30) | 0.013 |
| | AVG | **45** | (16.67) | **1** | (2.10) | **44** | (19.38) | |
| | Perm-catheter (Hickmann catheter) | **25** | (9.26) | **5** | (10.64) | **20** | (8.97) | |
| Local symptoms post vaccination (day 2) | | | | | | | | |
| Any symptom | | **104** | (38.52) | **26** | (55.32) | **78** | (34.98) | 0.009 |
| Induration | | **42** | (15.56) | **10** | (21.28) | **32** | (14.35) | 0.234 |
| Itch | | **11** | (4.07) | **3** | (6.38) | **8** | (3.59) | 0.412[b] |
| Redness | | **13** | (4.81) | **2** | (4.26) | **11** | (4.93) | 1.000[b] |
| Tenderness | | **78** | (28.89) | **19** | (40.43) | **59** | (26.46) | 0.055 |
| Warmth | | **8** | (2.96) | **4** | (8.51) | **4** | (1.79) | 0.033[b] |
| Swelling | | **41** | (15.19) | **9** | (19.15) | **32** | (14.35) | 0.380[b] |
| Local pain | | **80** | (29.63) | **19** | (40.43) | **61** | (27.35) | 0.075 |
| Systemic symptoms post vaccination (day 2) | | | | | | | | |
| Any symptom | | **168** | (62.22) | **37** | (78.72) | **131** | (58.74) | 0.010 |
| Feverishness (>37˚C and <38˚C) | | **54** | (20.00) | **17** | (36.17) | **37** | (16.59) | 0.002 |
| Low appetite | | **34** | (12.59) | **9** | (19.15) | **25** | (11.21) | 0.136[b] |
| Chills | | **41** | (15.19) | **13** | (27.66) | **28** | (12.56) | 0.009 |
| Fatigue | | **111** | (41.11) | **25** | (53.19) | **86** | (38.57) | 0.064 |
| Headache | | **50** | (18.52) | **18** | (38.30) | **32** | (14.35) | <0.001 |
| Muscle ache | | **51** | (18.89) | **10** | (21.28) | **41** | (18.39) | 0.682 |
| Joint pain | | **37** | (13.70) | **9** | (19.15) | **28** | (12.56) | 0.245[b] |
| Lethargy | | **54** | (20.00) | **11** | (23.40) | **43** | (19.28) | 0.521 |
| Insomnia | | **24** | (8.89) | **4** | (8.51) | **20** | (8.97) | 1.000[b] |
| Nausea or vomiting | | **17** | (6.30) | **3** | (6.38) | **14** | (6.28) | 1.000[b] |
| Diarrhea | | **10** | (3.70) | **5** | (10.64) | **5** | (2.24) | 0.017[b] |
| Abdominal pain | | **5** | (1.85) | **2** | (4.26) | **3** | (1.35) | 0.210[b] |
| Dyspnea | | **4** | (1.48) | **0** | (0.00) | **4** | (1.79) | 1.000[b] |
| Chest pain | | **8** | (2.96) | **0** | (0.00) | **8** | (3.59) | 0.358[b] |

*(Continued)*

**Table 1.** (Continued)

| Baseline characteristics and adverse effects 2 days after vaccination | | Overall | | Age (years) | | | | P-value |
|---|---|---|---|---|---|---|---|---|
| | | | | ≤ 55 (n = 47) | | >55 (n = 223) | | |
| Thirsty | | 51 | (18.89) | 10 | (21.28) | 41 | (18.39) | 0.682 |
| Fever (≥38˚C) | | 40 | (14.81) | 14 | (29.79) | 26 | (11.66) | 0.003 |
| Any fever (>37˚C) | | 65 | (24.07) | 21 | (44.68) | 44 | (19.73) | <0.001 |

Data are presented as mean (standard deviation) or n (%). ˚C, Celsius degree. [a] Student's t-test. [b] Exact test.

P-values were calculated using chi-square test, unless otherwise specified.

Kt/V, dialysis efficiency; CVD, cardiovascular disease; COPD, chronic obstructive pulmonary disease; AVF, arteriovenous fistula; AVG, arteriovenous graft.

Patients' age (mean ± SD) was 66.4 ± 11.5 and 82.6% (n = 223) were older than 55 years. BMI (mean ± SD) was 23.3± 3.59 and 64.1% were lower than 24. In the study group, 55.19% had DM and 67.78% had CVD. Two hundred (74.1%), 45 (16.7%), and 25 (9.26%) patients used arteriovenous fistula (AVF), arteriovenous graft (AVG), and Perm-catheter as long-term dialysis vascular access, respectively. Patients aged > 55 years were more likely to have DM (58.3%), CVD (69.96%), COPD (17.94%), malnutrition (albumin level 3.71 g/dL), and a higher proportion of AVG and Perm-catheter for dialysis vascular access (19.38% vs. 2.1%, p < 0.05) than those ≤ 55 years.

## Local and specific systemic adverse effects of ChAdOx1 nCoV-19 vaccination in our ESRD study cohort

Among the 270 participants, 104 had one or more local AEs, and 168 had one or more systemic AEs (Fig 1A). The severity and incidence of local and systemic AEs were the highest on day 2 after vaccination (p < 0.001). Local pain, tenderness, and induration around the injection site were the most frequent local AEs (Fig 1B). Fatigue, feverishness and lethargy were the most frequent systemic AEs (Fig 1C). In addition, 18.9% of patients had an unusual thirst. Younger ESRD patients have high rates of AEs, including both local and systemic AEs (both p < 0.05) (Table 1). The rates of at least one local and systemic AE were 55.32% and 78.72% in those ≤ 55 years and, 34.98% and 58.74% in those >55 years, respectively. The AEs included feverishness, chills, headache, diarrhea, and fever were significantly higher in the younger group than older group (all p < 0.05).

## Risk factors of specific local and systemic adverse effects of ChAdOx1 nCoV-19 (Oxford-AstraZeneca) vaccination in our ESRD study cohort

Age ≤55 years, female-sex, DM, and a higher platelet count were independent predictors for some local or systemic AEs after vaccination (Table 2). HD patients ≤55 years old were more likely to encounter systemic AEs, including feverishness (OR: 2.78, 95% CI: 1.43–5.88), chills (OR: 2.56, 95% CI: 1.11–5.88), fatigue (OR: 1.96, 95% CI: 0.99–3.85, p = 0.052), headache (OR: 3.85, 95% CI: 1.75–8.33), diarrhea (OR: 6.25, 95% CI: 1.45–25), fever (OR: 3.13, 95% CI: 1.37–7.14), and any fever (OR: 3.03, 95% CI: 1.45–5.88). Women had more systemic AEs than men, including nausea or vomiting (OR: 4, 95% CI: 1.11–14.28) and thirst (OR: 2.7, 95% CI: 1.25–5.88). Patients with DM had a 1.74- and 2.15-times higher rate of fatigue and joint pain, respectively, than those without DM. Individuals with a higher platelet count (an increase of every $10^4$/uL) were more likely to report local pain and chills.

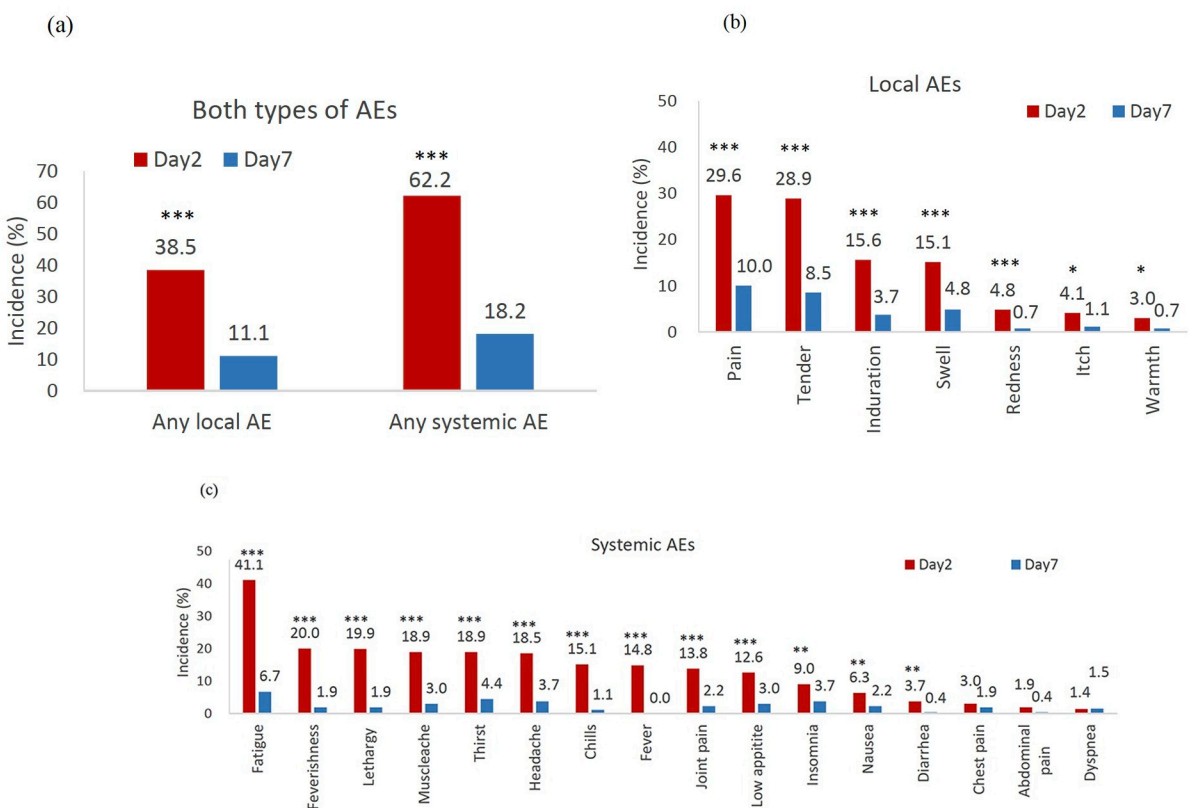

**Fig 1.** Incidence of any adverse effects (a), specific local symptoms (b), and specific systemic symptoms (c) on day 2 and day 7 of ChAdOx1 nCoV-19 (Oxford-AstraZeneca) vaccination (first dose) in ESRD patients. AEs, adverse effects. Incidence of AEs were compared between day 2 and day 7 using McNemar's test. ***, p < 0.001; **, 0.001 ≤ p < 0.01; *, 0.01 ≤ p < 0.05.

## Case reports of emergency department visit or hospitalization within 30 days after ChAdOx1 nCoV-19 vaccination in our ESRD study cohort

Twenty-nine (10.7%) patients visited the ED or were hospitalized within 30 days after vaccination. Two nephrologists evaluated the correlation between the clinical scenario and vaccine, referenced by medical charts and published reports, and excluded other relevant causes.

One patient suffered from herpes zoster (HZ) in the right leg five days after vaccination. Case reports on HZ emergence post mRNA and inactivated COVID-19 vaccines were recently published [15–17]. Therefore, a potential causal link between the events should be suspected. Another patient developed fever and chills on day 2 after vaccination. After thorough assessment, we excluded the possibility of infection and considered that his fever and chills were post-vaccination reactions.

To investigate the association between vaccination and increased risk of VAT, we further performed a self-controlled case series (SCCS) analysis in our study cohort. The observation period was initiated 6 months before and terminated 1 month after the first dose of vaccination. The observation period was grouped into 0–3 months prior to vaccination (pre1), 3–6 months prior to vaccination (pre2), and 1 month following vaccination (post). Compared with the post-period, the IRR-person ratio was 0.56 and 0.78 (IRR-event ratio was 0.78 and 1.00) in pre1 and pre2, respectively (Table 3). Our patients seemed to be less likely to suffer from VAT pre-vaccination than post-vaccination, although the difference in IR did not reach statistical significance.

**Table 2. Multivariable logistic regression analyses for risk factors of local (2a) and specific systemic (2b) adverse effects on day 2 of ChAdOx1 nCoV-19 (Oxford-AstraZeneca) vaccination (first dose) in ESRD patients.**

| Baseline characteristics | | (a) Local symptoms | | | | | | | | | | | | | | |
| --- | --- | --- | --- | --- | --- | --- | --- | --- | --- | --- | --- | --- | --- | --- | --- | --- |
| | | Induration | | | Itch | | | Redness | | | Tenderness | | | Swelling | | |
| OR | 95% CI | OR | 95% CI | P | OR | 95% CI | P | OR | 95% CI | P | OR | 95% CI | P | OR | 95% CI | P |
| Age (years) | >55 vs. ≤55 | 0.67 | 0.29 1.58 | 0.360 | 0.44 | 0.09 2.07 | 0.295 | 1.12 | 0.21 5.86 | 0.895 | 0.53 | 0.26 1.07 | 0.077 | 0.72 | 0.30 1.74 | 0.469 |
| Sex | Male vs. Female | 0.69 | 0.30 1.56 | 0.372 | 0.51 | 0.11 2.29 | 0.378 | 0.71 | 0.18 2.84 | 0.631 | 0.55 | 0.28 1.08 | 0.080 | 0.63 | 0.28 1.45 | 0.277 |
| BMI (kg/m$^2$) | ≥24 vs <24 | 0.97 | 0.43 2.18 | 0.945 | 2.57 | 0.57 11.58 | 0.218 | 1.94 | 0.49 7.57 | 0.343 | 1.19 | 0.62 2.29 | 0.604 | 1.16 | 0.51 2.60 | 0.728 |
| Diabetes | with vs. without | 0.80 | 0.40 1.61 | 0.531 | 2.03 | 0.47 8.74 | 0.340 | 1.09 | 0.33 3.66 | 0.888 | 1.31 | 0.73 2.35 | 0.359 | 0.88 | 0.43 1.81 | 0.736 |
| Albumin | 1 g/dL increase | 1.62 | 0.61 4.29 | 0.330 | 0.97 | 0.16 5.95 | 0.971 | 3.29 | 0.48 22.51 | 0.225 | 1.59 | 0.72 3.50 | 0.250 | 1.06 | 0.42 2.70 | 0.906 |
| kt/V | 1unit increase | 0.62 | 0.16 2.41 | 0.494 | 0.63 | 0.05 7.56 | 0.715 | 2.26 | 0.23 21.87 | 0.482 | 0.70 | 0.23 2.11 | 0.527 | 0.58 | 0.15 2.29 | 0.439 |
| Platelet | 10$^4$/uL increase | 1.00 | 0.94 1.05 | 0.896 | 1.01 | 0.91 1.12 | 0.853 | 1.01 | 0.92 1.10 | 0.874 | 1.04 | 0.99 1.08 | 0.114 | 1.00 | 0.95 1.06 | 0.985 |

| Baseline characteristics | | (a) Local symptoms | | | (b) Systemic symptoms | | | | | | | | | | | |
| --- | --- | --- | --- | --- | --- | --- | --- | --- | --- | --- | --- | --- | --- | --- | --- | --- |
| | | Local pain | | | Feverishness (>37°C and <38°C) | | | Low appetite | | | Chills | | | Fatigue | | |
| | | OR | 95% CI | P | OR | 95% CI | P | OR | 95% CI | P | OR | 95% CI | P | OR | 95% CI | P |
| Age (years) | >55 vs. ≤55 | 0.54 | 0.26 1.11 | 0.092 | 0.36 | 0.17 0.77 | 0.008 | 0.57 | 0.23 1.40 | 0.219 | 0.39 | 0.17 0.90 | 0.027 | 0.51 | 0.26 1.01 | 0.052 |
| Sex | Male vs. Female | 0.55 | 0.28 1.08 | 0.082 | 0.92 | 0.43 1.97 | 0.831 | 0.91 | 0.37 2.26 | 0.845 | 0.53 | 0.23 1.26 | 0.153 | 0.93 | 0.50 1.72 | 0.822 |
| BMI (kg/m$^2$) | ≥24 vs <24 | 1.41 | 0.73 2.72 | 0.310 | 1.15 | 0.55 2.41 | 0.705 | 1.03 | 0.43 2.49 | 0.949 | 0.58 | 0.25 1.39 | 0.223 | 1.24 | 0.68 2.26 | 0.491 |
| Diabetes | with vs. without | 1.09 | 0.61 1.95 | 0.766 | 1.15 | 0.60 2.19 | 0.683 | 0.94 | 0.44 2.03 | 0.879 | 1.86 | 0.87 3.96 | 0.109 | 1.74 | 1.02 2.98 | 0.041 |
| Albumin | 1 g/dL increase | 1.02 | 0.48 2.19 | 0.951 | 1.45 | 0.60 3.52 | 0.414 | 1.17 | 0.42 3.25 | 0.770 | 1.46 | 0.55 3.86 | 0.447 | 1.22 | 0.61 2.45 | 0.576 |
| kt/V | 1unit increase | 0.92 | 0.31 2.76 | 0.884 | 0.92 | 0.27 3.15 | 0.897 | 0.84 | 0.19 3.65 | 0.818 | 0.30 | 0.07 1.25 | 0.098 | 1.07 | 0.39 2.94 | 0.896 |
| Platelet | 10$^4$/uL increase | 1.06 | 1.01 1.11 | 0.011 | 1.01 | 0.96 1.06 | 0.754 | 1.01 | 0.95 1.07 | 0.695 | 1.06 | 1.00 1.12 | 0.050 | 1.01 | 0.97 1.05 | 0.629 |

| Baseline characteristics | | (b) Systemic symptoms | | | | | | | | | | | | | | |
| --- | --- | --- | --- | --- | --- | --- | --- | --- | --- | --- | --- | --- | --- | --- | --- | --- |
| | | Headache | | | Muscle ache | | | Joint pain | | | Lethargy | | | Insomnia | | |
| | | OR | 95% CI | P | OR | 95% CI | P | OR | 95% CI | P | OR | 95% CI | P | OR | 95% CI | P |
| Age (years) | >55 vs. ≤55 | 0.26 | 0.12 0.57 | 0.001 | 0.70 | 0.30 1.62 | 0.401 | 0.54 | 0.21 1.35 | 0.184 | 0.81 | 0.36 1.84 | 0.613 | 1.09 | 0.32 3.69 | 0.890 |
| Sex | Male vs. Female | 0.74 | 0.33 1.65 | 0.466 | 0.55 | 0.26 1.17 | 0.121 | 0.49 | 0.21 1.18 | 0.112 | 0.58 | 0.28 1.24 | 0.159 | 0.91 | 0.32 2.64 | 0.866 |
| BMI (kg/m$^2$) | ≥24 vs <24 | 0.65 | 0.29 1.45 | 0.293 | 1.49 | 0.70 3.17 | 0.300 | 1.33 | 0.56 3.13 | 0.517 | 1.25 | 0.61 2.57 | 0.548 | 1.82 | 0.65 5.10 | 0.258 |
| Diabetes | with vs. without | 1.17 | 0.59 2.30 | 0.652 | 1.44 | 0.73 2.82 | 0.293 | 2.15 | 0.96 4.83 | 0.063 | 1.22 | 0.63 2.34 | 0.560 | 1.08 | 0.42 2.77 | 0.876 |
| Albumin | 1 g/dL increase | 2.29 | 0.87 6.07 | 0.095 | 1.14 | 0.47 2.81 | 0.772 | 1.42 | 0.50 4.09 | 0.513 | 1.50 | 0.62 3.63 | 0.370 | 0.51 | 0.17 1.54 | 0.232 |
| kt/V | 1unit increase | 1.18 | 0.33 4.23 | 0.804 | 1.31 | 0.37 4.63 | 0.670 | 0.57 | 0.13 2.47 | 0.454 | 0.41 | 0.12 1.46 | 0.169 | 0.67 | 0.12 3.90 | 0.655 |
| Platelet | 10$^4$/uL increase | 1.02 | 0.97 1.08 | 0.397 | 1.02 | 0.97 1.08 | 0.367 | 1.04 | 0.98 1.10 | 0.251 | 1.00 | 0.95 1.05 | 0.939 | 1.02 | 0.95 1.09 | 0.649 |

| Baseline characteristics | | (b) Systemic symptoms | | | | | | | | | | | | | | |
| --- | --- | --- | --- | --- | --- | --- | --- | --- | --- | --- | --- | --- | --- | --- | --- | --- |
| | | Nausea or vomiting | | | Diarrhea | | | Thirst | | | Fever (≥38°C) | | | Any fever (>37°C) | | |
| | | OR | 95% CI | P | OR | 95% CI | P | OR | 95% CI | P | OR | 95% CI | P | OR | 95% CI | P |
| Age (years) | >55 vs. ≤55 | 0.86 | 0.21 3.50 | 0.835 | 0.16 | 0.04 0.69 | 0.014 | 0.70 | 0.30 1.64 | 0.410 | 0.32 | 0.14 0.73 | 0.007 | 0.34 | 0.17 0.69 | 0.003 |
| Sex | Male vs. Female | 0.25 | 0.07 0.90 | 0.034 | 0.44 | 0.09 2.22 | 0.319 | 0.37 | 0.17 0.80 | 0.012 | 1.61 | 0.66 3.96 | 0.298 | 1.15 | 0.56 2.38 | 0.700 |
| BMI (kg/m$^2$) | ≥24 vs <24 | 1.27 | 0.37 4.41 | 0.706 | 0.89 | 0.18 4.46 | 0.884 | 1.61 | 0.76 3.40 | 0.213 | 0.94 | 0.41 2.19 | 0.893 | 1.04 | 0.52 2.09 | 0.920 |
| Diabetes | with vs. without | 0.64 | 0.22 1.91 | 0.426 | 1.51 | 0.36 6.26 | 0.573 | 1.00 | 0.51 1.97 | 0.994 | 1.70 | 0.80 3.60 | 0.164 | 1.06 | 0.57 1.95 | 0.861 |
| Albumin | 1 g/dL increase | 0.70 | 0.19 2.67 | 0.604 | 0.49 | 0.10 2.31 | 0.365 | 1.01 | 0.42 2.42 | 0.987 | 1.30 | 0.48 3.53 | 0.601 | 1.42 | 0.62 3.25 | 0.412 |
| kt/V | 1unit increase | 0.53 | 0.07 3.97 | 0.533 | 0.36 | 0.03 4.46 | 0.426 | 0.53 | 0.15 1.90 | 0.330 | 1.19 | 0.29 4.85 | 0.807 | 0.99 | 0.31 3.17 | 0.987 |
| Platelet | 10$^4$/uL increase | 1.03 | 0.95 1.12 | 0.503 | 1.00 | 0.90 1.12 | 0.978 | 0.97 | 0.92 1.03 | 0.302 | 1.02 | 0.97 1.08 | 0.449 | 1.02 | 0.97 1.07 | 0.403 |

BMI, body mass index. OR, odd ratio. CI, confidence interval. P, p-value. Kt/v, dialysis efficiency. Any fever, raised body temperature ≥ 37°C.

## Discussion

We searched PubMed for articles published up to July 31, 2021, using the terms "COVID-19 vaccine", "adverse events", and "hemodialysis patients". Although the ChAdOx1 nCoV19

**Table 3. Number of patients, events and IRRs for vascular access thrombosis events before and after first dose vaccination with ChAdOx1.**

| Observations of vascular access thrombosis events | 3–6 m pre-vaccination (pre2) | 0–3 m pre-vaccination (pre1) | 1 m post-vaccination (post) |
|---|---|---|---|
| Person months (PMs) | 810 | 810 | 270 |
| Number of patients | 7 | 5 | 3 |
| IR, person/PMs | 0.009 | 0.006 | 0.011 |
| IRR | 0.78 (0.20–2.93) | 0.56 (0.13–2.33) | ref. |
| p-value | 0.714 | 0.422 | |
| Number of events | 9 | 7 | 3 |
| IR, events/PMs | 0.011 | 0.009 | 0.011 |
| IRR | 1.00 (0.20–4.93) | 0.78 (0.15–4.10) | ref. |
| p-value | 1.00 | 0.77 | |

IR, incidence rate. IRR, incidence rate ratio. Ref. reference period.

vaccine has the greatest global reach, no study has comprehensively evaluated its AEs in ESRD patients on HD. We investigated AEs during the 30-day follow-up period after the first vaccination with ChAdOx1 nCoV19 in ESRD patients on HD. At least one local AE was observed in 38.5% of patients; local pain, tenderness, and induration around the injection site were most common. At least one systemic AE was observed in 62.2%; fatigue, feverishness, and lethargy were the common. Younger age, female-sex, DM, and a higher platelet count increased the risk for some AEs. Moreover, our cohort is the first to describe post-vaccination thirst, which affected 18.9% of the participants with female predominance. According to the SCCS results, only a slightly increased risk of VAT was observed following vaccination but without statistical significance. To the best of our knowledge, this study is the first to evaluate AEs after vaccination with ChAdOx1 nCoV19 in ESRD patients on HD.

ESRD patients have distinct characteristics, including older age, malnutrition, and multiple comorbidities [18–20]. Our ESRD cohort had similar findings with 82.6% of patients older than 55 years. Those > 55 years had lower serum albumin levels, more comorbidities and non-AVFs (AVGs and Perm-Catheter) as long-term dialysis vascular access than those ≤ 55 years. In our ESRD cohort, younger individuals had more AEs than the older, which supports results of earlier studies in the general population [21–24]. However, the occurrence of muscle ache in our study was lower and revealed no difference in frequency between the two age groups. We speculate that this finding might result from the ubiquity of muscle wasting, malnutrition, and protein-energy wasting in the HD population [25].

As far as we know, our ESRD cohort is the first to report the post-vaccination thirst with 18.9% having unusual thirst with extreme water craving after vaccination and female predominance. Some patients even complained of increased salt appetite. The mechanism of post-vaccination thirst is unknown. We proposed it might be related to high plasma angiotensin II levels and some aggravating factors, such as hypovolemic thirst and restrained water intake [26–29]. Although many HD patients receive ACE inhibitors/ARBs for hypertension control, they hold the drugs on dialysis days to prevent intradialytic hypotension and inadequate dialysis. Hence, we suggest that the impacts of ACE inhibitors/ARBs on plasma angiotensin II levels were trivial in thirsty patients. However, further studies are needed to evaluate this aspect.

In our ESRD cohort, age ≤55 years, female-sex, DM, and a higher platelet count had a risk contribution to some AEs. Younger age and female sex have been reported as independent predictors of systemic AEs in the general population [21–24]. Recently, studies regarding age-related immunogenicity of COVID-19 vaccines echoed our results, showing that serum

neutralization, binding IgG or IgA levels, and T cell responses were more robust in younger adults after the first dose [22,30,31]. Furthermore, based on historical observations, females typically have higher antibody responses and experience more AEs following vaccination than males. These differences are documented in three various vaccines, including the measles, yellow fever, and influenza [32]. The mechanisms of sex differences in vaccine-induced immunity have involved immunological, hormonal, genetic, environmental, nutritional, and microbiota differences across males and females [33,34].

In contrast to age and gender, risk predictors of DM and serum platelet count have not been previously reported. In our study, individuals with DM had an increased risk of fatigue and joint pain. Metabolic reprogramming in DM drives the activation of immune cells and pro-inflammatory cytokines, leading to systemic inflammation and frailty [35–37]. Diabetic milieu combined with frailty might play a role in vaccine-related systemic AEs in HD patients [38]. On the other hand, we noted an interesting association between platelet count and the frequency of AEs. Beyond the traditional views of platelets as cells responsible for hemostasis and thrombosis, increasing evidence reveals that they are essential in diverse immunological responses, including innate and adaptive immune responses [39,40]. We propose that the versatile role of platelets in the immune system may enhance immunogenicity after vaccination and thus induce more AEs. Nevertheless, further research is warranted.

Across our ESRD cohort, five patients experienced possible severe AEs. One patient had a fever of 38.1˚C 2 days after vaccination but recovered rapidly without hospitalization. One female patient suffered from HZ infection 5 days after vaccination and received antiviral treatment afterward. HZ reactivation after COVID-19 vaccination (mRNA and inactivated vaccine) has been reported in healthy adults and patients with autoimmune rheumatic diseases [15–17,41,42]. Immunomodulation created by the vaccine may cause varicella zoster virus to escape from the latent phase [15–17]. Further studies are needed to elucidate the definitive causality between COVID-19 vaccination and HZ reactivation in patients with ESRD on dialysis.

Vaccine-induced immune thrombotic thrombocytopenia (VITT), a severe and rare complication after the ChAdOx1 nCoV19 vaccine, has attracted extensive attention, with reports of thrombotic events in the cerebral vein, splanchnic vein, and pulmonary vein [43,44]. Dialysis vascular access is also a venous system [45]. However, it is not known whether ChAdOx1 nCoV19 vaccine is an aggravating factor for thrombus formation in vascular access system. In our ESRD cohort, three patients experienced VAT within 30 days after vaccination. The three cases had normal platelet counts and coagulation tests when visiting the emergency department, indicating the thrombotic mechanism of acute VAT was different from VITT. Although we found somewhat lower IRRs of VAT in the pre-vaccination period by using the SCCS analysis, the difference was not statistically significant, which might result from the limited sample size of our study. Therefore, further comprehensive studies, such as a multicenter-based, are needed to provide robust results and answer this important clinical question.

Several methodological limitations warrant cautious interpretations of our study results. First, the major limitation of our study is the absence of a suitable comparison population. The study setting was considerably restricted due to the pandemic situation. At that time, Taiwan confronted a serious vaccine shortage. Under the national policy, ChAdOx1 nCoV-19 was the only vaccine we could access for HD patients. The first-dose vaccination rate was 91% in our HD center. The main reasons for patients not receiving the vaccination were primarily due to acute illness with hospitalization, severe infection, or frailty, precluding them from being an adequate comparison population. In addition, the small proportion of the control group would make further investigation difficult. Nevertheless, we made efforts to eliminate this main limitation by performing an SCCS analysis. Second, since this is a retrospective study, it

is hard to examine the changes of antibody titers in the first place to evaluate the immunogenicity post-vaccination. Third, the retrospective nature and limited sample size mean it is not possible to establish the causality between vaccination and potential severe AEs.

## Conclusions

Some patients with on HD had systemic or local AEs after vaccination. Younger age, female-sex, DM, and a higher platelet level had risk contributions for some AEs. Our report confirmed the safety profile of ChAdOx1 nCoV-19 in HD patients. The SCCS analysis did not show a significantly increased risk of VAT events in the first month following vaccination. Apart from VAT, other vaccine-related AEs, irrespective of local or systemic symptoms, had minor clinical significance on safety issues. However, further coordinated, multi-center, or registry-based studies are needed to determine causality. Moreover, this evidence may help assess the risks and benefits of vaccination and reduce vaccine hesitancy in this special population.

## Acknowledgments

Dr. Chin-Li Lu analyzed the data.

## Author Contributions

**Conceptualization:** I-Ning Yang, Chin-Li Lu, Chih-Chiang Chien.

**Data curation:** I-Ning Yang, Yu-Chi Kuo, Li-Hwa Tsai, Kuan Chieh Tu, Chih-Chiang Chien.

**Formal analysis:** Chin-Li Lu, Chih-Chiang Chien.

**Investigation:** Chih-Chiang Chien.

**Methodology:** Chin-Li Lu.

**Resources:** Yu-Chi Kuo, Li-Hwa Tsai.

**Software:** Chin-Li Lu.

**Supervision:** Chih-Chiang Chien.

**Validation:** Chih-Chiang Chien.

**Writing – original draft:** I-Ning Yang, Chin-Li Lu, Chih-Chiang Chien.

**Writing – review & editing:** Hung-Jen Tang, Jhi-Joung Wang, Chih-Chiang Chien.

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
