## [Decision Letter · Decision Letter 0]

6 Jul 2022

PONE-D-22-14735Safety of ChAdOx1 nCoV-19 vaccination in patients with end-stage renal disease on hemodialysisPLOS ONE

Dear Dr. Chien, 

Thank you for submitting your manuscript to PLOS ONE. After careful consideration, we feel that it has merit but does not fully meet PLOS ONE’s publication criteria as it currently stands. Therefore, we invite you to submit a revised version of the manuscript that addresses the points raised during the review process.

We look forward to receiving your revised manuscript.

Kind regards,

Donovan Anthony McGrowder, PhD., MA., MSc

Academic Editor

PLOS ONE

Journal Requirements:

This study was supported by grants CMFHR11002 from Chi-Mei Hospital.

Dr. Chin-Li Lu analyzed the data. 

 No.

No.

6. PLOS requires an ORCID iD for the corresponding author in Editorial Manager on papers submitted after December 6th, 2016. Please ensure that you have an ORCID iD and that it is validated in Editorial Manager. To do this, go to ‘Update my Information’ (in the upper left-hand corner of the main menu), and click on the Fetch/Validate link next to the ORCID field. This will take you to the ORCID site and allow you to create a new iD or authenticate a pre-existing iD in Editorial Manager. Please see the following video for instructions on linking an ORCID iD to your Editorial Manager account: https://www.youtube.com/watch?v=_xcclfuvtxQ.

7. We note that you have included the phrase “data not shown” in your manuscript. Unfortunately, this does not meet our data sharing requirements. PLOS does not permit references to inaccessible data. We require that authors provide all relevant data within the paper, Supporting Information files, or in an acceptable, public repository. Please add a citation to support this phrase or upload the data that corresponds with these findings to a stable repository (such as Figshare or Dryad) and provide and URLs, DOIs, or accession numbers that may be used to access these data. Or, if the data are not a core part of the research being presented in your study, we ask that you remove the phrase that refers to these data.

Additional Editor Comments:

Dear Dr. Chien,

The manuscript was revised in accordance with the reviewers’ comments and is provisionally accepted pending final checks for formatting and technical requirements.

Regards,

Dr. Donovan McGrowder (Academic Editor)

<o:p></o:p>

Reviewers' comments:

Reviewer's Responses to Questions

**Comments to the Author**

1. Is the manuscript technically sound, and do the data support the conclusions?

Reviewer #1: No

Reviewer #2: Yes

2. Has the statistical analysis been performed appropriately and rigorously? 

Reviewer #1: Yes

Reviewer #2: No

3. Have the authors made all data underlying the findings in their manuscript fully available?

Reviewer #1: Yes

Reviewer #2: Yes

4. Is the manuscript presented in an intelligible fashion and written in standard English?

Reviewer #1: Yes

Reviewer #2: Yes

5. Review Comments to the Author

Reviewer #1: I-Ning Yang et al. investigated the AEs within 30 days after the first dose of ChAdOx1 nCoV19 in ESRD patients under hemodialysis. Compared with the UK community cohort, the ESRD cohort had a lower risk of local AEs, but a higher risk of systemic AEs. The self-controlled case study analysis revealed no association between vaccination and vascular access thrombosis. Although the analysis of AEs of ChAdOx1 nCoV19 in ESRD patients under hemodialysis is interesting, there are several concerns as follow.

Major comments

1. In lines 62-63, the authors should describe the mortality ration in the literatures.

2. In lines 67-70, the antibody titers for SARS-CoV-2 should be investigated in the current ESRD cohort.

3. In lines 114-117, the ethnic differences may not be excluded by using UK community cohort as control, the cohort of Taiwan community should be compared.

4. In line 176, the ESRD patients under hemodialysis frequently complain a thirst. How did the authors conclude it was due to the vaccination?

5. In lines 218-219, how did the authors consider five cases were considered as AEs of vaccination?

6. In lines 268, the patients with thirsty did not receive the ARBs or ACE inhibitors?

Minor comments

1. In line 42, IRR should be spelled out.

Reviewer #2: Nothing can be said in 270 cases to examine the safety of the vaccine. At least 3000 cases are required. It is nonsense to compare and discuss the data of 270 HD patients and the data of 345280 of the UK community cohort because the numbers are too different and the characteristics of the population are also quite different.

Thrombosis with thrombocytopenia syndrome (TTS) is known as a significant and serious AE of ChAdOx1 nCoV19. The incidence of TTS is from 10 to 20 per 1 million inoculations. It is extremely rare and similar to heparin-induced thrombocytopenia (HIT).

You may have picked up acute VAT in light of its implications for TTS. Since there was no difference in the frequency of acute VAT before and after vaccination, you concluded that the relationship with vaccination is low.

It can be said that the incidence of VAT is common and has never increased with vaccination. Did you evaluate the findings similar to HIT such as thrombocytopenia, abnormal coagulation / fibrinolysis test, anti-PF4 antibody in VAT cases?

The other minor AEs such as fever, fatigue, local pain were not so important for safety of the vaccine.

---

## [Author Response · Author response to Decision Letter 0]

21 Jul 2022

Reviewer 1:

Major comment 1.

In lines 62-63, the authors should describe the mortality ration in the literatures.

Response:

Thanks for your valuable suggestion. We have reviewed the reference articles and described the mortality ratio in the Introduction session: "The relative risk of mortality was 45.4 for in-center HD patients with COVID-19 as compared with the general population4. Historically, this specific population had a higher mortality rate after pandemic viral infections, such as influenza (OR: 1.26, 95% CI: 1.15-1.38)5." (Introduction section, paragraph 1, page 4, lines 60-63)

Major comment 2.

In lines 67-70, the antibody titers for SARS-CoV-2 should be investigated in the current ESRD cohort.

Response:

We fully agree that the analysis of antibody titers for SARS-CoV-2 is essential. However, the retrospective nature of our study makes it difficult to investigate the data in the beginning. We have revised the Discussion section by adding this sentence to the Limitations paragraph: "Second, since this is a retrospective study, it is hard to examine the changes of antibody titers in the first place to evaluate the immunogenicity post-vaccination."

(Discussion section, paragraph 10, page 18, lines 299-301)

Major comment 3.

In lines 114-117, the ethnic differences may not be excluded by using UK community cohort as control, the cohort of Taiwan community should be compared.

Response:

Indeed, using the UK cohort as control may lack comparability due to ethnic differences. The numbers were also too different between the two populations. Besides, the Taiwan community surveillance data collected by an adverse effect self-reporting App is exclusively accessible by Taiwan CDC. Therefore, we decided to leave out all descriptions and interpretations of the UK cohort comparison in the literature, including in the Abstract, Introduction, Methods, Results, and Discussion sections.

Major comment 4.

In line 176, the ESRD patients under hemodialysis frequently complain a thirst. How did the authors conclude it was due to the vaccination?

Response:

As you mentioned, HD patients frequently feel thirsty. However, many patients reported that the extent of thirst was strong and "unusual" after vaccination. A weird salty taste sustained a few days after the shot and faded day by day, which drove them to drink more water on non-HD days. Consequently, we included this post-vaccination symptom as AEs in our analysis.

Major comment 5.

In lines 218-219, how did the authors consider five cases were considered as AEs of vaccination?

Response:

Thanks for your question. To investigate the association between vaccination and increased risk of VAT, we further performed a self-controlled case series (SCCS) analysis in our study cohort. The SCCS analysis did not find an elevated risk of VAT at 1 month following vaccination. 

Therefore, we agree with your viewpoint. The rationale of the statement is not clear and appropriate here. We decided to omit the sentence in the Results section.

(Results section, paragraph 1, page 12, lines 191)

Major comment 6.

In lines 268, the patients with thirsty did not receive the ARBs or ACE inhibitors?

Response:

Although many HD patients receive ACE inhibitors/ARBs for hypertension control, they hold the drugs on dialysis days to prevent intradialytic hypotension and inadequate dialysis. Hence, we suggest that the impacts of ACE inhibitors/ARBs on plasma angiotensin II levels were trivial in thirsty patients. (Discussion section, lines 239-243)

Minor comment 1.

In line 42, IRR should be spelled out.

Response:

Thanks for your kind reminder. We have spelled out the abbreviation of IRR in the Abstract. (Abstract, paragraph 2, page 2, line 40)

Reviewer 2:

Comment 1.

Nothing can be said in 270 cases to examine the safety of the vaccine. At least 3000 cases are required. It is nonsense to compare and discuss the data of 270 HD patients and the data of 345280 of the UK community cohort because the numbers are too different and the characteristics of the population are also quite different.

Response:

Indeed, using the UK cohort as control may lack comparability due to ethnic differences. The numbers were also too different between the two populations. Besides, the Taiwan community surveillance data collected by an adverse effect self-reporting App is exclusively accessible by Taiwan CDC. Therefore, we decided to leave out all descriptions and interpretations of the UK cohort comparison in the literature, including in the Abstract, Introduction, Methods, Results, and Discussion sections.

Comment 2.

Thrombosis with thrombocytopenia syndrome (TTS) is known as a significant and serious AE of ChAdOx1 nCoV19. The incidence of TTS is from 10 to 20 per 1 million inoculations. It is extremely rare and similar to heparin-induced thrombocytopenia (HIT).

You may have picked up acute VAT in light of its implications for TTS. Since there was no difference in the frequency of acute VAT before and after vaccination, you concluded that the relationship with vaccination is low.

It can be said that the incidence of VAT is common and has never increased with vaccination. Did you evaluate the findings similar to HIT such as thrombocytopenia, abnormal coagulation / fibrinolysis test, anti-PF4 antibody in VAT cases?

Response:

We fully agree with you. VAT incidence is much more common than TTS. The three cases that suffered from VAT had normal platelet counts and coagulation tests after vaccination (data were shown below), implying the thrombotic mechanism is quite different between VAT and TTS. Since no thrombocytopenia occurred, we did not further check fibrinolysis tests and anti-PF4 antibodies. We have revised our Discussion section by adding this sentence: "The three cases had normal platelet counts and coagulation tests when visiting the emergency department, indicating the thrombotic mechanism of acute VAT was different from VITT. "

 Platelet count(150-400 103/uL) PT (9.4-12.5 sec) aPPT(26.0-38.0 sec)

Case 1. 237 10.6 (INR: 0.95) 35.9 (aPPT ratio: 1.09)

Case 2. 179 11.0 (INR: 0.98) 33.4 (aPPT ratio: 1.01)

Case 3. 168 10.3 (INR: 0.92) 29.2 (aPPT ratio: 0.88)

 (Discussion section, paragraph 8, page 18, lines 284-286)

Comment 3.

The other minor AEs such as fever, fatigue, local pain were not so important for safety of the vaccine.

Response:

Indeed, apart from VAT, other vaccine-related AEs, irrespective of local or systemic symptoms, had minor clinical significance on safety issues. We have revised the sentences in the abstract and the conclusion paragraph.

(Abstract, paragraph 3, page 3, lines 48-50; Conclusion, page 19, lines 315-316)

---

## [Editor Report · Decision Letter 1]

12 Aug 2022

Safety of ChAdOx1 nCoV-19 vaccination in patients with end-stage renal disease on hemodialysis

PONE-D-22-14735R1

Dear Dr. Chien, 

We’re pleased to inform you that your manuscript has been judged scientifically suitable for publication and will be formally accepted for publication once it meets all outstanding technical requirements.

Kind regards,

Donovan Anthony McGrowder, PhD., MA., MSc

Academic Editor

PLOS ONE

Dear Dr. Chien,<o:p></o:p>

The manuscript was revised in accordance with the reviewers’ comments and is provisionally accepted pending final checks for formatting and technical requirements.

Regards,

Dr. Donovan McGrowder (Academic Editor)<o:p></o:p>

---

## [Editor Report · Acceptance letter]

22 Aug 2022

PONE-D-22-14735R1 

Safety of ChAdOx1 nCoV-19 vaccination in patients with end-stage renal disease on hemodialysis 

Dear Dr. Chien:

I'm pleased to inform you that your manuscript has been deemed suitable for publication in PLOS ONE. Congratulations! Your manuscript is now with our production department. 

Kind regards, 

on behalf of

Dr. Donovan Anthony McGrowder 

Academic Editor

PLOS ONE